# Whole-Body Vibration Training on Oxidative Stress Markers, Irisin Levels, and Body Composition in Women with Fibromyalgia: A Randomized Controlled Trial

**DOI:** 10.3390/bioengineering10020260

**Published:** 2023-02-16

**Authors:** Jousielle Márcia dos Santos, Redha Taiar, Vanessa Gonçalves César Ribeiro, Vanessa Kelly da Silva Lage, Pedro Henrique Scheidt Figueiredo, Henrique Silveira Costa, Vanessa Pereira Lima, Borja Sañudo, Mário Bernardo-Filho, Danúbia da Cunha de Sá-Caputo, Marco Fabrício Dias Peixoto, Vanessa Amaral Mendonça, Amandine Rapin, Ana Cristina Rodrigues Lacerda

**Affiliations:** 1Multicenter Graduate Program in Physiological Sciences, Brazilian Society of Physiology, Diamantina 39100-000, Brazil; 2Graduate Program in Functional Performance and Rehabilitation, Federal University of the Jequitinhonha and Mucuri Valleys, Diamantina 39100-000, Brazil; 3MATériaux et Ingénierie Mécanique, Université de Reims Champagne Ardenne, 51100 Reims, France; 4Department of Physical Education and Sports, Universidad de Sevilla, 41013 Seville, Spain; 5Mechanical Vibration Laboratory and Integrative Practices, Department of Biophysics and Biometrics, Institute of Biology Roberto Alcântara Gomes and Piquet Carneiro Polyclinic, Rio de Janeiro State University, Rio de Janeiro 20950-003, Brazil; 6Graduate Program in Health Sciences, Federal University of the Jequitinhonha and Mucuri Valleys, Diamantina 39100-00, Brazil; 7Faculté de Médecine, Université de Reims Champagne Ardennes, UR 3797 VieFra, 51097 Reims, France

**Keywords:** chronic condition, fibromyalgia, obesity, visceral adipose tissue, oxidative stress, muscle pain, whole-body vibration

## Abstract

(1) Background: Mitochondrial dysfunction and redox imbalance seem to be involved in fibromyalgia (FM) pathogenesis. The results of our previous studies suggest that whole-body vibration training (WBVT) would improve redox status markers, increase blood irisin levels, and ameliorate the body composition of women with FM. (2) Objective: The current study aimed to investigate WBVT on oxidative stress markers, plasma irisin levels, and body composition in women with FM. (3) Methods: Forty women with FM were randomized into WBVT or untrained (UN) groups. Before and after 6 weeks of WBVT, body composition was assessed by dual-energy radiological absorptiometry (DXA), and inflammatory marker activities were measured by enzymatic assay. (4) Results: Body composition, blood irisin levels, and oxidative stress markers were similar between UN and WBVT groups before the intervention. After 6 weeks of intervention, the WBVT group presented higher irisin levels (WBVT: 316.98 ± 109.24 mg·dL³, WBVT: 477.61 ± 267.92 mg·dL³, *p* = 0.01) and lower TBARS levels (UN: 0.39 ± 0.02 nmol MDA/mg protein, WBVT: 0.24 ± 0.06 nmol MDA/mg protein, *p* = 0.001) and visceral adipose tissue mass (UN: 1.37 ± 0.49 kg, WBVT: 0.69 ± 0.54 kg, *p* = 0.001) compared to the UN group. (5) Conclusions: Six weeks of WBVT improves blood redox status markers, increases irisin levels, and reduces visceral adipose tissue mass, favoring less cell damage and more outstanding oxidative balance in women with FM.

## 1. Introduction

Fibromyalgia (FM) is characterized by widespread non-inflammatory musculoskeletal pain and tenderness in at least 11 of 18 tender sites for a minimum period of three months [1]. FM affects 1–2% of people, and 89% of those affected are women. In addition, this disease can present with a variety of symptoms, such as fatigue, anxiety, depression, unrefreshing sleep, muscle stiffness, or irritable bowel syndrome [2]. Although the exact causes of FM are still unknown, its pathophysiology seems to be influenced by mitochondrial malfunction and redox imbalance [3]. Reactive oxygen and nitrogen species (ERONs) are produced in significant amounts as a result of oxidative stress, which is a significant event in the pathophysiology of FM [4,5]. ERONs are highly reactive chemical species created by catalyzing transition metals such as iron, copper, or manganese. They have an unpaired electron. Due to their altered quantity of mi-matched valence electrons, these hazardous compounds are highly reactive when they are first formed. In rheumatologic disorders such rheumatoid arthritis (RA), ankylosing spondylitis (AS), and chronic fatigue syndrome (CFS), these ERONs may be crucial [6,7]. The extent of oxidative stress in FM is poorly understood, but malondialdehyde levels are higher and superoxide dismutase levels are lower than in controls [8,9].

The main symptoms of FM (pain, stiffness, and fatigue) are related to skeletal muscle dysfunction; thus, evaluations of muscle alterations and mitochondrial function in women with FM have been a major focus of investigation. In recent years, the identification of new myokines and adipokines has expanded the understanding of the neurohumoral functions of skeletal muscle and adipose tissue [10,11,12]. Irisin has emerged as a crucial hormone involved in a number of exercise-induced benefits, including improvements in body composition and muscular performance [13]. Recent evidence has found that irisin has a pivotal role in redox status balance by reducing ROS production and protecting cells against oxidative stress damage [14].

Therapies frequently focus on symptom treatment because the pathogenesis of FM is unclear [15]. For FM patients, whole-body vibration training (WBVT) has recently been suggested as a time-effective, risk-free, feasible, and well-tolerated exercise therapy [16]. In healthy people, a single WBV exercise effectively increases circulating irisin levels [17]. In addition, data indicate that WBVT increases irisin levels in individuals with chronic obstructive pulmonary disease who are hospitalized [18]. A single WBV session improved the redox balance [19] and increased levels of the anti-inflammatory myokines STNFr1, STNFr2, and IL6 [20] in women with FM, according to earlier research from our lab.

We can point out that that studies that involved training with WBVT for 6 weeks show that WBVT encourages higher BDNF levels, with concomitant improvement in lower limb muscle strength, aerobic capacity, clinical symptoms, and quality of life in women with FM [21]. In contrast to exercise alone, Alerton-Geli and colleagues showed that 6-weeks of a traditional exercise program with supplemental WBV reduces pain and fatigue [22]. Another study investigated the effectiveness of a 6-week exercise program in WBVT regarding the improvement of strength and health status in women with FM. The findings revealed that FM women benefited from participating in a traditional exercise program of 6 weeks with supplemental WBV [23]. Additionally, 4 weeks of WBVT was shown to be useful in relieving FM symptoms [24]. In a recent study conducted by our team, it was found that FM women’s aerobic capacity, clinical symptoms, strength, and quality of life all improved after 6 weeks of WBVT [21]. These findings support the idea that training with WBVT for six weeks has a positive impact on the clinical aspects of FM.

Short-term WBVT’s impact on oxidative stress indicators, irisin levels, and body composition in FM patients is still unknown. We hypothesized that short-term WBVT would enhance redox status, increase irisin levels, and improve the body composition of women with FM based on the findings of our previous studies. Therefore, the purpose of this study was to investigate whether 6 weeks of WBVT modified oxidative stress markers, irisin blood levels, and body composition in FM women.

## 2. Materials and Methods

### 2.1. Ethical Considerations

The Federal University of the Jequitinhonha and Mucuri Valleys Ethics and Research Committee approved this study (number 4.510.517). All participants received information on the study’s aims, methods, possible risks, discomforts, and benefits. Each participant provided a written informed consent and was allowed to leave the study at any time.

### 2.2. Study Design

This randomized controlled clinical study with disguised allocation and blinding of the assessor for score counts was developed at the Universidade Federal dos Vales do Jequitinhonha e Mucuri (Diamantina/Minas Gerais, Brazil). Participants were recruited by convenience from health centers in the local community between June 2017 and June 2018.

Individual allocation codes placed inside sealed, opaque envelopes were used for randomization. The study was registered in the Brazilian Clinical Trials Registry (REBEC: RBR-2x84bbn). This study adhered to the Consolidated Standards of Reporting Studies (CONSORT) and Standard Protocol Items: Recommendations for Interventional Trials (SPIRIT).

### 2.3. Inclusion Criteria

The inclusion criteria were perimenopausal or menopausal women with FM diagnosis confirmed by a clinical rheumatologist; aged 50 to 60 years; non-smokers; non-consumers of alcohol; and insufficiently physically active in the last 24 months, according to the American College of Sports Medicine (ACSM) classification [25].

### 2.4. Exclusion Criteria

Acute hernias, orthopedic and prosthetic lesions, metabolic or neuromuscular diseases, epilepsy, stroke, and patients taking oral or topically applied immunosuppressive medications were among the possible contraindications to the stimulus with WBV excluded from the study. Additionally, patients receiving psychiatric follow-ups were also excluded (corticosteroids).

Before the baseline evaluation and allocation, participants were checked for eligibility requirements. All participants were divided into groups by a single researcher. The trained group (WBVT), which underwent WBVT, and the untrained group (UN), which remained inactive, were assigned to the participants randomly. All participants familiarized with the WBV stimulus and obtained instructions on the proper exercise technique. The UN group’s participants received weekly phone instructions to maintain their daily pattern of activities.

### 2.5. Participants

Initially, 71 people were evaluated for eligibility: 28 did not match the requirements for inclusion and 3 declined to take part. As a result, 40 FM women took part in the trial and were then randomly allocated to either the WBVT or the UN group (Figure 1).

### 2.6. Evaluations

On the first visit to the laboratory (LAFIEX-UFVJM), participants arrived at 7 a.m. after fasting for 8 to 10 h. Personal sociodemographic data, medical history, and information on lifestyle were gathered. Immediately after the collection of personal information and 48 h after the last exercise session, the evaluations were performed in the same sequence. Blood was collected, and immediately after blood collection participants were given a standard snack. A specialized professional performed the blood collection. Then, body mass was recorded to an accuracy of 0.1 kg and height to an accuracy of 0.5 cm using a scale and stadiometer, model 110 (Welmy, São Paulo, Brazil), and the body mass index (BMI) was calculated as body mass (kg)/height^2^ (m). Thereafter, body composition was evaluated by DXA. All participants were instructed to maintain their usual activities and dietary routines during the study period.

### 2.7. Intervention

For six weeks, the group training underwent WBVT three times each week on alternate days. The exercise routine involved dynamic squats on a synchronous vibrating platform (FitVibe^®^ Excel Pro, Gymna Uniphy, Belgium) for three seconds up and three seconds down. Over the course of six weeks, the number of sets gradually increased (6 to 8 sets). The subject had to do a semi-complete knee extension, or up, for 3 s (angle 10°), followed by 3 s of knee flexion, or down, (angle 60°), for each squat. Each squat required 8 s because the participants were asked to hold the final angles of 10° and 60° for one second. The patient was told to take a 30 s rest between sets on the vibratory platform with the motor turned off. Around 180 to 624 s, or 3 to 10.6 min, were spent on each workout session [26]. The participants were also told to keep their feet on the platform and their heads, arms, and spine in the proper positions (simulating the motion of sitting in a chair) to reduce resonant catastrophe [27]. Patients who performed the exercises barefoot [28] avoided the dampening impact caused by various shoes. Additionally, a predefined distance between the feet was set, 14 cm to the right and 14 cm to the left of the platform’s vibration center [29], to guarantee each lower limb received an equal amount of vibration stimuli (Figure 2). Additionally, a fixed distance was used to ensure that each lower limb received the same amount of vibration stimuli (Figure 2). The mechanical stimulation parameters of the vibration were frequency of 35–40 Hz, amplitude of 4 mm, and acceleration gravity ranging from 2.78 to 3.26 g (Table 1).

The training protocol was adapted from previous studies by our research group [21,26]. A physical therapist monitored all training sessions, including pain intensity and perceived exertion (RPE), before and after each intervention session. Of note, we did not find any difference in RPE (study internal control) between participants of WBVT during all periods of WBVT. The procedures in this study were reported according to the Consensus Statement from an International Group of Experts [30].

Primary outcomes: Oxidative stress measurements and circulating irisin levels.

### 2.8. Oxidative Stress Biomarkers

The median cubital vein was punctured aseptically to obtain blood samples stored in tubes containing EDTA anticoagulant and was centrifuged at 3000× *g* for 10 min at 20 °C to remove cells and debris. Plasma and erythrocyte were stored at −80 °C. Oxidative stress biomarkers were evaluated by determining plasma levels of lipid peroxidation products through the reactive substances with thiobarbituric acid (TBARS) [31]. Superoxide dismutase (SOD) [32] and catalase (CAT) [33] activities, as well as total antioxidant capacity of non-enzymatic antioxidants of plasma (FRAP) [34], were measured by enzymatic assays according to previous studies. The TBARS levels were reported in nanomoles of MDA per milligram of protein. The SOD activity was reported in units (U) per milligram of protein, and the CAT activity was reported by D.E./min per milligram of protein, where D.E. represents the variation in enzyme activity for 1 min. Total antioxidant capacity was reported as micrograms of FeSO_4_ per milligram of protein.

### 2.9. Irisin Plasma Levels

According to the manufacturer’s instructions, plasma irisin levels were determined using a traditional sandwich enzyme-linked immunosorbent test kit (DuoSet, R&D Systems, Minneapolis, MN, USA). Detection limit was 5.0 pg/mL for the kit.

### 2.10. Secondary Outcomes: Body Composition—Determination of Body Composition

Dual-energy X-ray absorptiometry: A densitometer (Lunar Radiation Corporation, Madison, Wisconsin, USA, model DPX) was used to determine the lean body mass [35]. The appendicular lean body mass (ALM) was obtained as the sum of the muscle mass of the four members divided by height squared (Baumgartner index), representing the lean body mass [35]. Body fat mass was obtained by dividing total fat mass by height squared [36]. Visceral adipose tissue (VAT) obtained by the mass value determined by examination (Figure 3).

### 2.11. Statistical Analysis

Data were reported as mean and standard deviation. Data normality was verified by the Shapiro–Wilk test. An independent sample t-test was used to compare each dependent variable between groups at baseline (pre-training).

A randomized block design was used for each group (model: yik = μ + mi + pk + eik; where μ refers to mean; yik refers to the observation of patient (block) k at the time i; mi refers to effect of the moment i; pk refers to effect of the patient (block) k; and eik refers to experimental error).

For the ANOVA, the following model was considered: yijk = μ +mi + gj + mgij + pk(j) + eik(j), where μ refers to mean of the two groups; yijk refers to observation of patient (block) k at the time i within group j; mi refers to effect of the moment i; gj refers to effect of group j; mgij refers to effect of the interaction between mi and gj; pk(j) refers to the effect of patient k of group j; and eik(j) refers to mean experimental error.

All hypothesis tests (F and post hoc) were performed considering ANOVA. For the effects of groups, moments, and interactions, the F test was used once it was conclusive. For comparisons involving combinations of groups and moments, the Scheffé test was used. All tests were performed at 5% probability.

An effect size (eta squared: ƞ^2^) < 0.25 represented a small effect, 0.25–0.4 a moderate effect, and >0.4 a large effect [37]. No intention-to-treat analyses were conducted because there were no missing data. The GPower software version 3.1.9.2 (Kiel University, Quiel, Alemanha), was used to calculate the sample size. Taking into account the comparison of the blood oxidative stress levels between groups (the primary outcome), the sample size was estimated in 40 volunteers (20 per group), with an effect size of 1.1, the statistical power of 80%, and an alpha error of 5%. [34]. The Statistical Package GraphPad Prism version 5.00 was used for building graphics and statistical analyses.

## 3. Results

At baseline, there was no significant difference in clinical, anthropometric, and blood biomarker characteristics between the groups. Participants in the UN and WBVT groups were obese (grade 1) according to the body mass index (BMI) classification (30.86 ± 3.98 and 30.26 ± 4.92 kg/m^2^, respectively) (Table 2).

In the follow-up, there was a decrease in VAT for WBVT in relation to UN (*p* = 0.001). In addition, there was an increase in blood levels of irisin for WBVT compared to UN (*p* = 0.009). After 6 weeks of the WBVT protocol, CAT activity increased in both groups (*p* = 0.006) with between-group (*p* = 0.002) and interaction effects (*p* = 0.01).

Although there were no significant differences between the groups in blood levels of FRAP, there was an interaction effect (*p* = 0.01). In addition, although post hoc analysis revealed no between-group or interaction effects, a pooled analysis revealed within-group differences in blood levels of TBARS in the WBVT group. SOD, lean mass index, and fat mass index were similar at baseline, and there was no interaction or difference between-groups at follow-up (Table 3) (Figure 4).

## 4. Discussion

The current study looked into how 6 weeks of WBVT affected oxidative stress markers, irisin levels, and body composition in FM women. The major results showed that WBVT (i) reduced VAT, (ii) increased blood irisin levels, and (iii) reduced blood levels of TBARS. To our knowledge, this study is the first to investigate the effect of WBVT on oxidative stress markers and irisin blood levels in women with FM.

Physical inactivity is frequently linked to an increase in FM symptoms, which exacerbates the disease’s effects and makes it more difficult for patients to perform daily tasks, leading to obesity and further reductions in physical capability and quality of life [38]. Women with FM frequently have obesity [39,40]. The prevalence of obesity in women with FM was 78% in the current study, and the mean BMI was 30.86 kg/m^2^. The prevalence of obesity and overweight in FM patients has also been noted in other studies, and these conditions are linked to the disease’s severity, increasing symptoms such increased pain, fatigue, poor sleep, and a higher incidence of mood disorders [41,42]. Consequently, weight loss is a useful strategy for reducing symptoms. Studies have shown that non-pharmacological treatments such as physical activity and nutrition are effective treatment options for FM [43,44,45,46,47,48,49,50]. They also show that weight management, modified diets with a high antioxidant content, nutritional supplementation, and physical activity are all helpful in reducing symptoms.

Studies have found a cross talk between body fat, lean mass, and neuroinflammation [51]. Obesity and dynapenia are often harmful to skeletal muscle function and can lead to low-grade systemic inflammation, especially in chronic diseases such as FM [52]. In addition to having a better ability to diminish visceral adipose tissue in obese middle-aged people, a prior study indicated that chronic exposure to WBVT lowers adipogenesis [53]. Additionally, WBVT improves strength and muscle mass [50], as well lowers blood glucose, triglycerides, and cholesterol [54]. After 6 weeks of WBVT in our study, we could see a decrease in VTA in women with FM. It is feasible to suggest that VTA reductions with WBVT are possible based on these prior findings and those of our investigation.

It is recognized that weight loss enhances FM patients’ quality of life, sleep, and pain level [17]. Studies have emphasized the value of short-term investigations that establish a causal connection between WBVT, body composition, and clinical FM aspects. Exercise seems to be able to trigger a transient response in both biological systems [29]. Although excessive ROS might be deleterious, it is now understood that transitory alterations brought on by exercise are essential for promoting adaptability [38]. For the treatment of FM, it is advised that patients obtain patient education, cognitive-behavioral therapy, medication, healthy sleeping patterns, and combined exercise. Exercise specifically benefits FM patients’ physical fitness, aerobic capacity, pain management, and quality of life [23]. We can assume that WBVT was beneficial in body composition parameters, redox status markers, and irisin levels in women with FM, and these positive effects of WBVT may contribute to the management of clinical FM symptoms given the importance of appropriate exercise strategies to maintain the integrity of clinical and physical aspects in FM [55,56].

Skeletal muscle secretes the hormone irisin in response to physical exercise; this hormone has been linked to several positive effects on energy metabolism. Irisin acts as a messenger in white adipose tissue, modifying its phenotype to beige adipocyte, enhancing adipocyte thermogenic capacity [57,58]. Evidence points to an association between irisin secretion rising with an improvement in glucose and insulin metabolism. Our data (Table 3) showed that WBVT increases the plasma irisin levels in women with FM [59]. Recent studies have suggested that irisin has the potential to treat chronic diseases due to antioxidative effects. Irisin may reduce ROS generation and improve antioxidant defenses throughout different downstream molecules such as UCP-2 and GPX4. These effects are especially important in chronic diseases where ROS generation may be enhanced and antioxidant defenses diminished [14].

The role of mitochondrial oxidative stress in FM pathogenesis has been widely discussed [6,9]. Previous studies have shown that patients with FM have high plasma levels of TBARS [4,5] and low plasma levels of SOD and CAT [7,8,9], indicating a pro-oxidative state. Among the complementary strategies for treating FM, physical exercise is recommended.

It has been extensively discussed [6,9] how mitochondrial oxidative stress contributes to FM development. A pro-oxidative state is evident in individuals with FM, as demonstrated by their high plasma levels of TBARS [4,5] and low plasma levels of SOD and CAT [7,8,9]. Exercise is one of the additional methods advised for treating FM. However, it is important to note that depending on the level of physical exertion, gender, age, and physical condition, improper exercise might have a harmful effect on the body. Due to increased creation of free radicals during exercise, the detrimental consequences of physical activity are frequently related to an imbalance between levels of antioxidants (both low molecular weight antioxidants and antioxidant enzymes) and reactive oxygen and nitrogen species. Redox balance appears to be restored by exercise [60]. We emphasize that WBVT is a method of brief, moderately intense exercise that has gained recent recognition as a promising adjuvant therapy for chronic disorders [19,20,21,24,38]. In this study, we discovered that 6 weeks of WBVT in FM women increased blood irisin levels and decreased TBARS and VAT levels, balancing out the effects in favor of less cellular damage and greater oxidative balance.

## 5. Conclusions

In FM women, WBVT reduces VAT, raises blood irisin levels, and improves blood redox status markers. Taken together, our data suggest that WBVT is a time-effective and promising adjuvant therapy for FM patients.

This study is registered in the Brazilian Clinical Trials Registry (REBEC; RBR-2x84bbn) https://ensaiosclinicos.gov.br/rg/RBR-2x84bbn, accessed on 28 april 2022.

## 6. Practical Application

WBVT is an alternative non-pharmacological treatment for FM that improves patients’ physical, clinical, functional, and inflammatory conditions. Given that increased visceral adipose tissue, low irisin levels, and systemic imbalance redox are all modifiable biological variables associated with chronic low-grade inflammation, WBVT may be a promising therapy for future non-pharmacological therapeutic approaches. As a result, the findings presented in this study can be used to establish novel methods for planning studies to understand the effects of WBVT on the modulation of inflammatory markers as well as the clinical, physical, and functional aspects of FM women.

## 7. Strengths and limitations

This study has some limitations. The patients’ self-reported dietary intake and physical activity levels could be a study limitation. We used a short-term protocol of 6 weeks; thus, the more long-term effects of WBVT deserve future investigations. Moreover, our results cannot be generalized to all patients with FM (e.g., men and non-obese patients).

This study also has strengths. We highlight the methodological quality of our investigation as a strength, since it scored an 8 out of 10 on the Physiotherapy Evidence Database Scale (PEDro scale) [61]. It is worth noting that the items that did not allow the total score of the present study were blinding the subjects and the therapist who applied the therapy. However, these items do not apply to WBVT studies. Additionally, our analyses showed a large effect size (greater than 0.8) for most outcomes [37]. Thus, the sample size was adequate for the experimental design of this study, and the results were clinically important. Finally, we underline the use of DXA, a gold-standard method for assessing body composition, as well as the evaluation of irisin levels and oxidative stress biomarkers, which have not been previously investigated in studies involving WBVT as an intervention in women with FM.

## Figures and Tables

**Figure 1 bioengineering-10-00260-f001:**
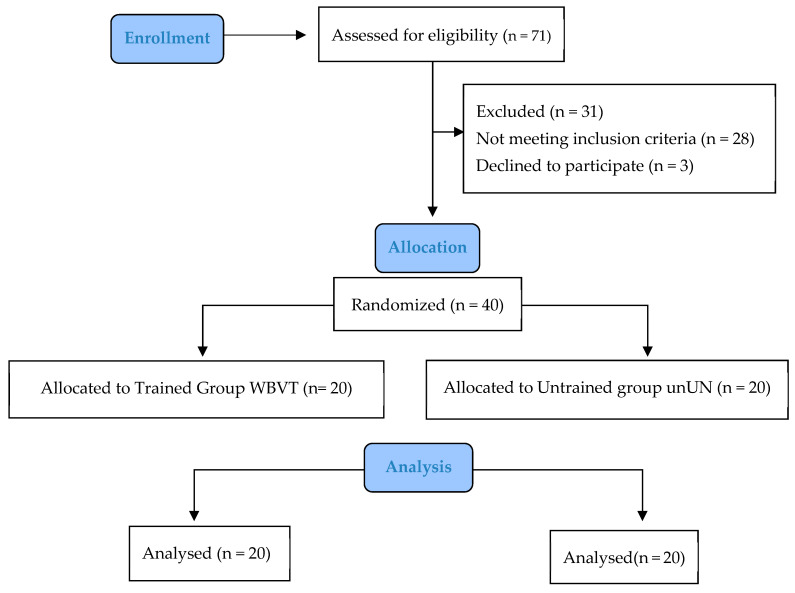
Study flowchart showing recruitment and final analysis of participants.

**Figure 2 bioengineering-10-00260-f002:**
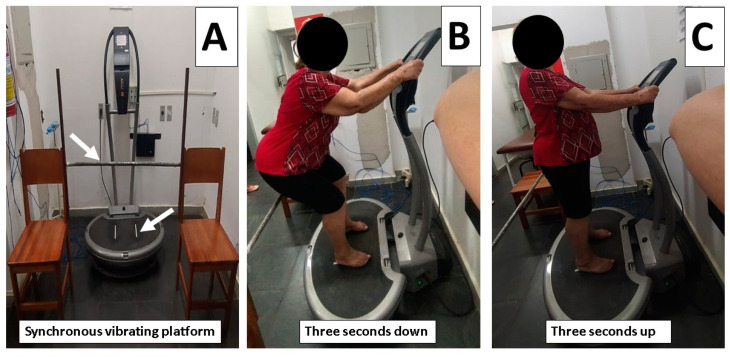
The whole-body vibration training (WBVT). Synchronous vibrating platform and positioning during experimental conditions. (**A**) Synchronous vibrating platform, arrows indicating distance 14 cm to the right and 14 cm to the left of the platform’s vibration center and fixed distance to ensure that each lower limb received the same amount of vibration stimuli; (**B**) flexion, or down, for each squat for 3 s (angle 60°); (**C**) extension, or up, for 3 s (angle 10°).

**Figure 3 bioengineering-10-00260-f003:**
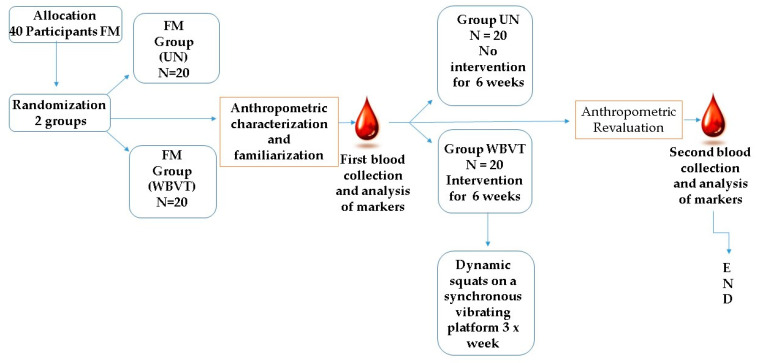
Diagram illustrating the study’s design. Design of the study; fibromyalgia (FM), untrained (UN), whole-body vibration training (WBVT).

**Figure 4 bioengineering-10-00260-f004:**
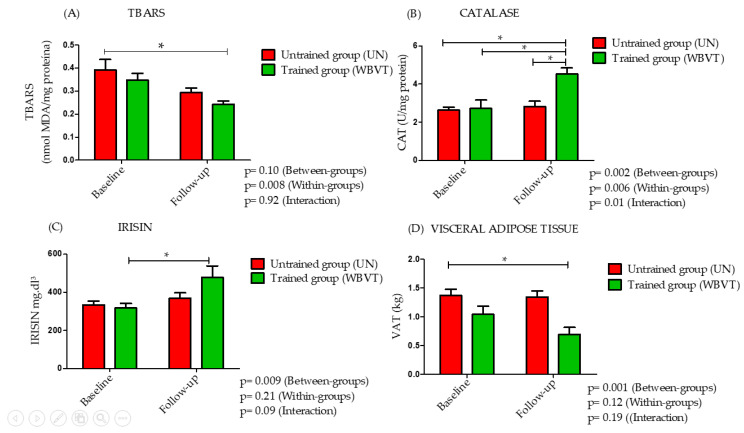
Effects of 6 weeks of whole-body vibration training (WBVT) stimulus on blood oxidative stress biomarkers and irisin levels. Plasma levels of thiobarbituric acid (TBARS) (**A**), catalase (CAT) (**B**), irisin (**C**), and visceral adipose tissue (VAT) (**D**) in control group (UN; n = 20) and training group (WBVT; n = 20). Values are presented as means and standard deviation. * Difference between groups (*p* < 0.05).

**Table 1 bioengineering-10-00260-t001:** Whole-body vibration training load program (adapted from [27]).

	Vibration Parameters (Intervention)						
Weeks	Frequency (Hz)	Amplitude (mm)	Acceleration (g)	Total Time per Set	Number of Repetitions per Set	Total Number of Sets	Rest Time between Sets
1	35	4	2.78	16 s	5	6	30 s
2	35	4	2.78	24 s	8	7	30 s
3	35	4	2.78	32 s	10	8	30 s
4	40	4	3.26	35 s	11	8	30 s
5	40	4	3.26	40 s	13	8	30 s
6	40	4	3.26	48 s	16	8	30 s

Hz: hertz, mm: millimeters, G: gravity acceleration.

**Table 2 bioengineering-10-00260-t002:** Clinical, anthropometric, and blood biomarker characteristics of participants at baseline.

Characteristics	UN	WBVT	*p* Value
Age (years)	54.31 ± 7.62	55.12 ± 6.44	0.71
Time from diagnosis (years)	8.15 ± 2.37	8.40 ± 2.68	0.75
BMI (kg/m^2^)	30.86 ± 3.98	30.26 ± 4.92	0.67
Lean mass index (kg)	8.31 ± 1.04	8.18 ± 0.76	0.67
Fat mass index (kg)	15.99 ± 1.24	15.93 ± 2.08	0.92
VAT (kg)	1.37 ± 0.48	1.34 ± 0.48	0.37
TBARS (nmol MDA/mg protein)	0.39 ± 0.20	0.34 ± 0.13	0.40
FRAP (FeSO4.1^−1^mgprotein^−1^)	8.34 ± 2.37	6.87 ± 2.40	0.98
CAT (U/mg protein)	2.64 ± 0.71	2.72 ± 2.02	0.86
SOD (U/mg protein)	0.30 ± 0.16	0.33 ± 0.24	0.59
Irisin (ng/mL)	333.88 ± 90.01	316.98 ± 109.24	0.59

Values are presented as means and standard deviation. UN: untrained group (n = 20); WBVT: whole-body vibration training (n = 20); BMI: body mass index; VAT: visceral adipose tissue mass; TBARS: thiobarbituric acid reactive substances; FRAP: ferric reducing antioxidant power; CAT: catalase; SOD: superoxide dismutase; *p* value: unpaired *t*-test.

**Table 3 bioengineering-10-00260-t003:** Effects of 6 weeks of WBVT on blood oxidative stress biomarkers and irisin levels.

Outcomes			Between Groups			Within Groups			Interaction		
Variables	Groups	Follow-Up	p1	F	η^2^	p2	F	η^2^	p3	F	η^2^
TBARS (nmol MDA/mg protein)	UN WBVT	0.29 ± 0.09 0.24 ± 0.06 *	0.10	2.69	0.73	0.008	12.17	0.92	0.92	0.01	0.10
FRAP (FeSO4.1^−^.mg protein^−1^)	UN WBVT	7.24 ± 1.53 8.33 ± 2.99	0.74	0.11	0.10	0.72	0.13	0.12	0.01	5.78	0.85
CAT (U/mg protein)	UN. WBVT	4.08 ± 1.52 *^#^ 2.83 + 1.21	0.002	9.96	0.50	0.006	7.85	0.44	0.01	6.46	0.40
SOD (U/mg protein)	UN. WBVT	0.31 ± 0.14 0.36 ± 0.20	0.36	0.95	0.48	0.68	0.19	0.15	0.87	0.03	0.10
Irisin (ng/mL)	UN. WBVT	368.65 ± 125.70 477.62 ± 267.93 ^#^	0.009	7.10	0.88	0.21	1.57	0.61	0.09	2.95	0.74
VAT (kg)	UN. WBVT	1.04 ± 0.62 ^#^ 0.69 ± 0.12	0.001	16.53	0.94	0.12	2.45	0.71	0.19	1.75	0.63
Lean mass index	UN. WBVT	8.20 ± 1.00 8.60 ± 0.67	0.41	0.65	0.39	0.48	0.48	0.32	0.18	1.74	0.63
Fat mass index	UN. WBVT	16.24 ± 1.22 15.78 ± 1.98	0.90	0.18	0.13	0.50	0.46	0.31	0.60	0.28	0.22

Values are presented as means and standard deviation. WBVT: whole-body vibration; TBARS: thiobarbituric acid reactive substances; FRAP: ferric reducing antioxidant power; CAT: catalase; SOD: superoxide dismutase; VAT: visceral adipose tissue mass. Bold values represent statistical difference. Two-way ANOVA with two complete block designs (between groups, within groups, interaction analyses); F values; η^2^: eta partial; UN: untrained group (n = 20); IG: trained group (n = 20). * Statistical difference within groups (WBVT follow-up × WBVT baseline). # Statistical difference between groups (WBVT follow-up × UN follow-up)**.**

## Data Availability

The study data are with the researchers and can be provided when necessary.

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
