# Peer review of "Whole-Body Vibration Training on Oxidative Stress Markers, Irisin Levels, and Body Composition in Women with Fibromyalgia: A Randomized Controlled Trial"

_bioengineering, 2023, doi:10.3390/bioengineering10020260_

Round 1

Reviewer 1 Report

It is an original study with an adequate methodology and very interesting results, easy to read and very well structured.

It is important, however, in the methodological aspect, to record schematically the research design.

It is necessary to add the DOI to each bibliographic reference or, failing that, the URL

Author Response

Reviewer 1

It is an original study with an adequate methodology and very interesting results, easy to read and very well structured.

Comment 1: It is important, however, in the methodological aspect, to record schematically the research design.

Answer: Thank you for this comment. Your suggestion was incorporated into the manuscript. Text modified: (Figure 2, Page 7, line 238).

Comment 2: It is necessary to add the DOI to each bibliographic reference or, failing that, the URL.

Answer: We added the DOI to each bibliographic reference. Text modified: (Page 14, line 465 the 623).

Reviewer 2 Report

Title: “Whole-body vibration training on oxidative stress markers, irisin levels, and body composition in women with fibromyalgia: A randomized controlled trial “

This work aimed at investigating the effect of Whole-Body Vibration Training (WBVT) on oxidative stress markers, plasma irisin levels, and body composition in women with fibromyalgia (FM). In this work forty women with FM were randomized into WBVT (TR) or Untrained (UN) groups. In particular, before and after 6 weeks of WBVT, body composition was assessed by dual-energy radiological absorptiometry (DXA), while inflammatory markers activities were measured by enzymatic assay. The authors showed that, while before the intervention, body composition, blood irisin levels and oxidative stress markers were similar between UN and TR groups, after 6 weeks of WBVT, TR group presented higher irisin levels (p=0.01), lower TBARS levels (p=0.002) and visceral adipose tissue mass (p= 0.001) with reference to the UN group. As a consequence, the authors claim that six weeks of WBVT improves blood redox status markers, increases irisin levels and reduces visceral adipose tissue mass, this could result in less cell damage and greater oxidative balance in women with FM.

General comment: Although the aim of this work is interesting, the current version should be reworked to increase its quality and impact. The manuscript is quite short and the general impression is that some experimental details could be better described, enlarging the main text and providing more plots and clear details about experimental trials. The WBVT procedure should be better described and introduced for interested readers. All the (few) tables should be coupled with plots showing the statistical significance of the main results. Overall it is not clear why a period of only 6 weeks has been chosen, and what are the known (or assessed) results of the application of the WBVT for longer or shorter periods. Indeed, the authors claim that they would investigate the “long-tem” effect of WBVT. In addition, one of the final claims of the authors is that “it is plausible that these positive effects of WBVT could contribute to the management of FM's clinical symptoms. [52,53]”, too weak to convince the readers of the value of this work.

Some detailed comments:

lines: “However, the effectiveness of long-term WBVT on oxidative stress markers, 71
irisin levels, and body composition in FM patients remains unclear. In light of the results 72
of our previous studies, we hypothesized that long-term WBVT would improve redox 73
status; increase irisin levels and ameliorate body composition of women with FM. There- 74
fore, this study aimed to look into the effects of 6 weeks of WBVT on oxidative stress 75
markers, irisin blood levels, and body composition in women with FM. 7

*) A better description of the WBVT should be provided. The authors should better explain why a 6 weeks period has been chosen for experiments to investigate the “long terms” effects of WBVT.

Results” section

*) Please insert in this section only results and insert experimental details in “Materials and methods section” (e.g., “Initially, 71 individuals were screened for eligibility, 28 did not meet the inclusion criteria, and 3 refused to participate. As a result, 40 women with FM participated in the 1trial and were randomly assigned to one of two groups: the WBVT or the UN. (Figure 1). ,etc)

Figure 1. Flow diagram of the study
*) This figure should be improved together with its caption. It seems to belong to the “Material and Methods” section.

*) The statistical methods used and their sensitiveness should be better explained both in “Materials and Methods” for methodology and here for the achieved results.

Tables 2,3 should be improved as well as their captions. If possible, the authors should insert some addition plots to show in a clear and fast way what are the main difference about the two different groups.

Lines: “4. Discussion 229

The current study investigated the effects of 6-week WBVT on oxidative stress mark- 230

ers, irisin levels, and body composition in women with Fibromyalgia. The main findings 231

demonstrated that WBVT: (i) reduced VAT (ii) increased blood irisin levels; (iii) and re- 232

duced blood levels of TBARS. To our knowledge, this study is the first to investigate the 233

effect of WBVT on oxidative stress markers and irisin blood levels in women with FM. 234

Physical inactivity is frequently associated with an increase in FM symptoms, which 235

worsens the disease's effects and makes it harder for patients to carry out everyday tasks, 236

resulting in obesity and additional declines in physical capability and quality of life[35]. 237

Obesity is a common condition in women with FM [36,37]. In the current study, the prev- 238

alence of obesity in women with FM was 78% and the mean BMI was 30.86 kg/m2. Other 239

studies have also reported a high prevalence of obesity in women with FM[36-42] and 240

acknowledge that non-pharmacological therapies, such as physical exercise, are the most 241

effective strategies for weight management and adjuvant treatment of FM. 242

Previous studies demonstrated that WBVT improved balance and muscle strength 243

(20,43-45), reduced pain, and increased quality of life in patients with FM (19,20,46). Fur- 244

thermore, we also found an increase in blood levels of brain-derived neurotrophic factor 245

(BDNF) after WBVT compared to non-trained patients with FM (21). 246

Studies have found a cross talk between body fat, lean mass and neuroinflammation 247

[47]. Obesity and dynapenia are often harmful to skeletal muscle function and can lead to 248

low-grade systemic inflammation, especially in chronic diseases such as FM [48]. On the 249

other hand, when compared to conventional aerobic exercise, WBVT significantly reduces 250

VAT in obese patients [49]. Moreover, WBVT reduces blood cholesterol, triglycerides and 251

glucose [49,50], and increases strength and muscle mass [51]. 252

There is evidence that weight loss improves life, sleep quality, and reduces pain in 253

patients with FM [17]. Several studies have pointed out the importance of longitudinal 254

studies establishing the cause and effect relationship between WBVT, body composition 255

and clinical aspects of FM. Our study presents strong evidence of the beneficial effects of 256

WBVT on body composition, redox status markers, and irisin levels in women with FM. 257

Thus, it is plausible that these positive effects of WBVT could contribute to the manage- 258

ment of FM's clinical symptoms. [52,53]. 259

Skeletal muscle secrets the hormone irisin in response to physical exercise; this hor- 260

mone has been linked to several positive effects on energy metabolism. Irisin acts as a 261

messenger in white adipose tissue, modifying its phenotype to beige adipocyte, enhancing 262

adipocyte thermogenic capacity [54,55]. Evidence points to an association between irisin 263

secretion rising with an improvement of glucose and insulin metabolism. Our data (Table 264

3) showed that WBVT increases the plasma irisin levels in women with FM. Recent studies 265

have suggested that irisin has the potential to treat chronic diseases due to its antioxida- 266

tive effects. Irisin may reduce ROS generation and improve antioxidant defenses through- 267

out different downstream molecules such as UCP-2 and GPX4. These effects are espe- 268

cially important in chronic diseases where ROS generation may be enhanced and antiox- 269

idant defenses diminished. (14). 270

The role of mitochondrial oxidative stress in the pathogenesis of FM has been widely 271

discussed [6,9]. Previous studies showed that patients with FM present high plasma levels 272

of TBARS [4,5] and low plasma levels of SOD and CAT [7-9], indicating a pro-oxidative 273

status. Vigorous physical exercise may increase the generation of reactive oxygen species 274

in chronic disease conditions; however, when performed at moderate intensity, exercise 275

seems to restore the redox balance [56]. WBVT is a moderate-intensity and short-duration 276

exercise modality and has been recognized in recent years as a feasible adjuvant therapy 277

in chronic disease conditions [19-21,24,38]. In this study, we found that 6 weeks of WBVT 278

in women with fibromyalgia enhanced blood irisin levels and decreased TBARS levels 279

and VAT, towards the balance in favor of less cell damage and greater oxidative balance.”

*) In these lines the authors claim that “Several studies have pointed out the importance of longitudinal studies establishing the cause and effect relationship between WBVT, body composition and clinical aspects of FM. Our study presents strong evidence of the beneficial effects of WBVT on body composition, redox status markers, and irisin levels in women with FM. Thus, it is plausible that these positive effects of WBVT could contribute to the management of FM's clinical symptoms. [52,53].” However, they claim “a strong evidence of the beneficial effects of WBVT on body composition, redox status markers, and irisin levels in women with FM”, from side, while they claim only that “it is plausible that these positive effects of WBVT could contribute to the management of FM's clinical symptoms. [52,53]”. Therefore, the readers are not too convinced of the value of this work. Indeed, they should also demonstrate that a “strong evidence” exist between the increase of irisin levels a greater oxidative balance and clear beneficial effects on FM. Please, rework and improve this section accordingly.

Author Response

Reviewer 2

General comment: Although the aim of this work is interesting, the current version should be reworked to increase its quality and impact. The manuscript is quite short and the general impression is that some experimental details could be better described, enlarging the main text and providing more plots and clear details about experimental trials. The WBVT procedure should be better described and introduced for interested readers.

Comment 1: A better description of the WBVT should be provided. The authors should better explain why a 6 weeks period has been chosen for experiments to investigate the “long terms” effects of WBVT.

Answer: Previous studies using WBVT as an intervention for chronic condition also considered a period of 6 weeks, considered as short term period (see a list of references below), as effective to demonstrate improvements in clinical symptoms, functionality and inflammatory aspects in chronic conditions. Text modified: (Page 2, Paragraph: 4, Lines: 81 the 94).

References:

  1. Sañudo, Borja et al. “Effect of whole-body vibration exercise on balance in women with fibromyalgia syndrome: a randomized controlled trial.” Journal of alternative and complementary medicine (New York, N.Y.) 18,2 (2012): 158-64. doi:10.1089/acm.2010.0881
  2. Alentorn-Geli, Eduard et al. “Six weeks of whole-body vibration exercise improves pain and fatigue in women with fibromyalgia.” Journal of alternative and complementary medicine (New York, N.Y.) 14,8 (2008): 975-81. doi:10.1089/acm.2008.0050
  3. Sañudo, B et al. “The effect of 6-week exercise programme and whole body vibration on strength and quality of life in women with fibromyalgia: a randomised study.” Clinical and experimental rheumatology 28,6 Suppl 63 (2010): S40-5.
  4. Ribeiro, V G C et al. “Inflammatory biomarkers Answers after acute whole body vibration in fibromyalgia.” Brazilian journal of medical and biological research = Revista brasileira de pesquisas medicas e biologicas 51,4 e6775. 1 Mar. 2018, doi:10.1590/1414-431X20176775
  5. Ribeiro, Vanessa G C et al. “Efficacy of Whole-Body Vibration Training on Brain-Derived Neurotrophic Factor, Clinical and Functional Outcomes, and Quality of Life in Women with Fibromyalgia Syndrome: A Randomized Controlled Trial.” Journal of healthcare engineering 2021 7593802. 30 Nov. 2021, doi:10.1155/2021/7593802
  6. Alev, Alp et al. “Effects of whole body vibration therapy in pain, function and depression of the patients with fibromyalgia.” Complementary therapies in clinical practice 28 (2017): 200-203. doi:10.1016/j.ctcp.2017.06.008

Comment 2: Please insert in this section only result and insert experimental details in “Materials and methods section”. Figure 1. Flow diagram of the study (This figure should be improved together with its caption. It seems to belong to the “Material and Methods” section).

Answer: We agree with your suggestion. We modified “Materials and methods section” as suggested. Text modified: (Page 2, Paragraph: 4, Lines: 81 the 94).

Comment 3: The statistical methods used and their sensitiveness should be better explained both in “Materials and Methods” for methodology and here for the achieved results.

Answer: We included additional information about the statistical methods used in the article as follows.

“Data were reported as mean and standard deviation. Data normality was verified by the Shapiro–Wilk test. An independent sample t-test was used to compare each dependent variable between groups at baseline (pre-training).

A randomized block design was used for each group [model: yik = μ + mi + pk + eik; where μ refers to the mean; yik refers to the observation of patient (block) k at the time i; mi refers to the effect of the moment i; pk refers to the effect of the patient (block) k; and eik refers to experimental error].

For the ANOVA, the following model was considered: yijk = μ +mi + gj + mgij + pk(j) + eik(j), where μ refers to the mean of the two groups; yijk refers to the observation of patient (block) k at the time i within group j; mi refers to the effect of the moment i; gj refers to the effect of group j; mgij refers to the effect of the interaction between mi and gj; pk(j) refers to the effect of patient k of group j; and eik(j) refers to the mean experimental error.

All hypothesis tests (F and Post Hoc) were performed considering ANOVA. For the effects of groups, moments, and interactions, the F test was used once it is conclusive. For comparisons involving combinations of groups and moments, the Scheffé test was used. All tests were performed at 5% probability.

Effect size (eta squared: Æž2) < 0.25 represented a small effect, 0.25-0.4 a moderate effect, and > 0.4 a large effect [34]. No intention-to-treat analyses were conducted because no missing data. The GPower software version 3.1.9.2 was used to calculate the sample size. Taking into account the comparison of the blood oxidative stress levels between groups (the primary outcome), the sample size was estimated in 40 volunteers (20 per group), with an effect size of 1.1, the statistical power of 80%, and an alpha error of 5%. [34]. The Statistical Package GraphPad Prism version 5.00 was used for building graphics and statistical analyses.” Text modified: (Page 7, Paragraph: 1, Lines: 241 the 266).

Comment 4: Tables 2,3 should be improved as well as their captions. If possible, the authors should insert some addition plots to show in a clear and fast way what are the main difference about the two different groups.

Answer: We have updated the descriptions of tables 2, 3, and attached a graph that compares the results across groups.

Comment 5: In these lines the authors claim that “Several studies have pointed out the importance of longitudinal studies establishing the cause and effect relationship between WBVT, body composition and clinical aspects of FM. Our study presents strong evidence of the beneficial effects of WBVT on body composition, redox status markers, and irisin levels in women with FM. Thus, it is plausible that these positive effects of WBVT could contribute to the management of FM's clinical symptoms. [52,53].” However, they claim “a strong evidence of the beneficial effects of WBVT on body composition, redox status markers, and irisin levels in women with FM”, from side, while they claim only that “it is plausible that these positive effects of WBVT could contribute to the management of FM's clinical symptoms. [52,53]”. Therefore, the readers are not too convinced of the value of this work. Indeed, they should also demonstrate that a “strong evidence” exist between the increase of irisin levels a greater oxidative balance and clear beneficial effects on FM. Please, rework and improve this section accordingly.

Answer: We agree with the reviewer. We rewrote and improved this section accordingly. Text modified: (Page 11, Paragraph: 1, Lines: 347 the 353).

Reviewer 3 Report

The article entitled “Whole-body vibration training on oxidative stress markers, irisin  levels, and body composition in women with fibromyalgia: A randomized controlled trial” has been reviewed, and it seems relevant to too add knowledge regarding the fibromyalgia pathogenesis. However, the authors should improve some aspects, such as language (i.e. the same observed in papers published by the research group, inclusion of schematic figures in the methods section, improvements on the discussion. Consequently, that manuscript should be accepted after these corrections.

Specific Comments

Abstract

Line 33- 36 – Include average and standard deviation values as well as p-values.

Line 36 to 38 - Could the authors give a practical application to this study as well?

Introduction

Line 46 – Which studies? Quote them in this sentence.

Line 71 to 76 – Rewrite all these sentences. It does not make sense to the reader.

 3rd and 4th para should be reorganized in one.  

I recommend the authors to review the language of the text, because there are some repetitive sentences from others papers of the same group.

Methods

I suggest that authors review the language in the methods section as there are conflicts with other articles published by the same group.

Results

I suggest that the authors modify Table 2 by a graphic.

Discussion

Line 239 to 242

What about the dietary intake? Authors discussed only considering the physical exercise. We know that, both work together in the case of obese people.

Line 243 to  246

A brief explanation about this result should be included.

Authors should include the limitations of the present study.

Conclusion

A practical application should be included for people who do not have access to WBVT treatment.

Author Response

Reviewer 3

Comments and Suggestions for Authors

The article entitled “Whole-body vibration training on oxidative stress markers, irisin  levels, and body composition in women with fibromyalgia: A randomized controlled trial” has been reviewed, and it seems relevant to too add knowledge regarding the fibromyalgia pathogenesis. However, the authors should improve some aspects, such as language (i.e. the same observed in papers published by the research group, inclusion of schematic figures in the methods section, improvements on the discussion. Consequently, that manuscript should be accepted after these corrections.

Comment 1: Abstract

Line 33- 36 – Include average and standard deviation values as well as p-values.

Line 36 to 38 - Could the authors give a practical application to this study as well?

Answer: We agree with the reviewer and include mean and standard deviation values as well as p-values in the summary. Text modified: (Page 1,Lines: 36 the 40). In addition, we provided a section with practical application of the study. (Section inserted after conclusions).

Comment 2: Line 46 – Which studies? Quote them in this sentence.

Answer: We agree with the reviewer's observations and updated the reviewer's recommendation by including additional studies. Text modified: Page 2, Paragraph: 1, Line: 53 the 55).

Comment 3: 3rd and 4th para should be reorganized in one.  

Answer: We agree with the reviewer's opinion and changed it, combining the third and fourth into one. Text modified: (Page 2, Paragraph: 3, Lines: 72 the 80).

Comment 4: I suggest that authors review the language in the methods section as there are conflicts with other articles published by the same group.

Answer: It should be noted once the present work is an arm of a larger randomized clinical trial study (approved by the Federal University of the Jequitinhonha and Mucuri Valleys Ethics and Research Committee - number 4.510.517), some of the methods used in the current study share some similarities with those used in our previous study, "Efficacy of Whole-Body Vibration Training on Brain-Derived Neurotrophic Factor, Clinical and Functional Outcomes, and Quality of Life in Women with Fibromyalgia Syndrome: A Randomized Controlled Trial". However, the results of the current study are unpublished and include outcomes (such as anthropometric profile, oxidative biomarkers, and irisin levels) not covered in our earlier research.

References:

  1. Ribeiro, Vanessa G C et al. “Efficacy of Whole-Body Vibration Training on Brain-Derived Neurotrophic Factor, Clinical and Functional Outcomes, and Quality of Life in Women with Fibromyalgia Syndrome: A Randomized Controlled Trial.” Journal of healthcare engineering 2021 7593802. 30 Nov. 2021, doi:10.1155/2021/7593802

Comment 5: I suggest that the authors modify Table 2 by a graphic.

Answer: We agree with the reviewer that a graph comparing results between groups should be attached. Text modified: (Page 8, after Table 3).

Comment 6: Line 239 to 242

What about the dietary intake? Authors discussed only considering the physical exercise. We know that, both work together in the case of obese people.

Answer: We agree with the reviewer's observations and added studies on dietary intake and obesity. Text modified: (Page 10, Paragraph: 2, Lines: 337 the 345).

Comment 7: Line 243 to  246A brief explanation about this result should be included.

Answer: We agree with the reviewer and included a brief explanation. Text modified: (Page 11, Paragraph: 1, Lines: 349 the 355).

Comment 8: Authors should include the limitations of the present study.

Answer: The section on strengths and limitations highlights the limitations of the study. Text modified: (Page 12, Lines: 421 the 424).

Comment 9: A practical application should be included for people who do not have access to WBVT treatment.

Answer: We agree with the reviewer and provided a section with practical application of the study. (Section added following conclusions). Text modified: (Page 12, Lines: 409 the 416).

Round 2

Reviewer 2 Report

Title: “Whole-body vibration training on oxidative stress markers, irisin levels, and body composition in women with fibromyalgia: A randomized controlled trial “

This work aimed at investigating the effect of Whole-Body Vibration Training (WBVT) on oxidative stress markers, plasma irisin levels, and body composition in women with fibromyalgia (FM). In this work forty women with FM were randomized into WBVT (TR) or Untrained (UN) groups. In particular, before and after 6 weeks of WBVT, body composition was assessed by dual-energy radiological absorptiometry (DXA), while inflammatory markers activities were measured by enzymatic assay. The authors showed that, while before the intervention, body composition, blood irisin levels and oxidative stress markers were similar between UN and TR groups, after 6 weeks of WBVT, TR group presented higher irisin levels (p=0.01), lower TBARS levels (p=0.002) and visceral adipose tissue mass (p= 0.001) with reference to the UN group. As a consequence, the authors claim that six weeks of WBVT improves blood redox status markers, increases irisin levels and reduces visceral adipose tissue mass, this could result in less cell damage and greater oxidative balance in women with FM.

General comment: The authors revised their work. However, some minor points should be improved:

Figure 3.

Plots and caption”

Effects of 6-weeks of WBVT on blood oxidative stress biomarkers and irisin levels. Effect of whole-body vibration training (WBVT) stimulus on plasma levels of thiubarbituric acid (TBARS) (A), catalase (CAT) (B), 321
irisin (C) and visceral adipose tissue (VAT) (D) in control group (UN; n = 20) and training group (WBVT; n = 20). Values are pre- 322
sented as means and standard deviation. * difference between groups (p < 0.05).

*) All plots in Figure 3 should be reworked to allow the readers to clearly understand all the needed information. All the main labels should be made understandable increasing their size. All the plots should be provided in high resolution. Perhaps also different colours may be beneficial.

Author Response

Reviewer 2

General comment: The authors revised their work. However, some minor points should be improved:

 *) All plots in Figure 3 should be reworked to allow the readers to clearly understand all the needed information. All the main labels should be made understandable increasing their size. All the plots should be provided in high resolution. Perhaps also different colours may be beneficial.

Answer: We agree with the reviewer. We have made adjustments to figure 3. Text modified: (Page 10, after Table 3).

Reviewer 3 Report

The authors improved a lot the manuscript reads. However, some comments were not responded and included after the first review. Consequently, the manuscript should be accepted after these corrections. 

Specific comments

Methods 

A schematic figure from the experimental protocol (subjects, equipment, the WBVT device acting on untrained and trained subjects).

Regards., 

Author Response

Reviewer 3

General comment: Comments and Suggestions for Authors

The authors improved a lot the manuscript reads. However, some comments were not responded and included after the first review. Consequently, the manuscript should be accepted after these corrections.

A schematic figure from the experimental protocol (subjects, equipment, the WBVT device acting on untrained and trained subjects).

Answer: Thanks for the suggestion. We have attached a schematic figure from the experimental protocol.

Text modified: (Page 6, after Table 1).
